# A statistical framework for quantifying the nuclear export rate of influenza viral mRNAs

**Michi Miura[1]\*, Naho Kiuchi[1], Siu-Ying Lau[2], Bobo Wing-Yee Mok[2], Hiroshi Ushirogawa[1], Tadasuke Naito[1], Honglin Chen[2], Mineki Saito[1]\***

[1]Department of Microbiology, Kawasaki Medical School, Kurashiki, Japan; [2]State Key Laboratory for Emerging Infectious Diseases, Department of Microbiology, The University of Hong Kong, Hong Kong, China

## eLife Assessment

This **important** study combines virology experiments and mathematical modeling to determine the nuclear export rate of each of the eight RNA segments of the influenza A virus, leading to the proposal that a specific retention of mRNA within the nucleus delays the expression of antigenic viral proteins. The proposed model for explaining the differential rate of export is **compelling**, going beyond the state of the art, but the experimental setup is only in partial support and further studies will be needed to confirm the proposed mechanism.

**\*For correspondence:**
michi.miura.res@gmail.com
(MM);
mineki@med.kawasaki-m.ac.jp
(MS)

**Abstract** Influenza A virus transcribes viral mRNAs from the eight segmented viral genome when it infects. The kinetics of viral transcription, nuclear export of viral transcripts, and their potential variation between the eight segments are poorly characterised. Here, we introduce a statistical framework for estimating the nuclear export rate of each segment from a snapshot of *in situ* mRNA localisation. This exploits the cell-to-cell variation at a single time point observed by an imaging-based *in situ* transcriptome assay. Using our model, we revealed the variation in the mRNA nuclear export rate of the eight viral segments. Notably, the two influenza viral antigens hemagglutinin and neuraminidase were the slowest segments in the nuclear export, suggesting the possibility that influenza A virus uses the nuclear retention of viral transcripts to delay the expression of antigenic molecules. Our framework presented in this study can be widely used for investigating the nuclear retention of nascent transcripts produced in a transcription burst.

## Introduction

Temporal gene expression is crucial for viral pathogens to proliferate in the host. Human exogenous retroviruses reverse-transcribe their RNA genome, insert it into the host chromatin, and use host factors to minimise the viral protein production to evade the immune response and persist in the host (*Bangham et al., 2019*). In contrast, influenza A virus infects the respiratory tract and produces thousands of viral progenitors in a single round of infection to spread in the surrounding epithelial cells by transcribing and replicating its RNA genome (*Einav et al., 2020*). Influenza A virus genome is segmented into eight single-stranded RNAs, and expression of each gene is essential for productive infection (*Dou et al., 2018*). Unlike the human exogenous retroviruses, influenza A virus does not seem to have a sophisticated mechanism of controlling the viral gene expression at the transcription level (*Walker and Fodor, 2019*), such as transcription factor binding or epigenetic modifications. However, it has been known for decades that hemagglutinin (HA) and neuraminidase (NA), the two

influenza viral antigens targeted by the host immune response, are produced late during the course of infection (*Lamb and Choppin, 1983*; *Shapiro et al., 1987*; *Hatada et al., 1989*), a mechanism of which remains currently unclear.

The eight viral RNA segments are transcribed by the same set of molecular machinery (*Te Velthuis and Fodor, 2016*); thus, the mechanism of influenza viral transcription does not seem to explain the temporal protein expression. The RNA-dependent RNA polymerase of influenza A virus, comprising polymerase basic 1 (PB1), polymerase basic 2 (PB2), and polymerase acidic (PA) subunits, is responsible for viral transcription. These subunits are associated with the partially hybridised 5′ and 3′ joint end of a viral genomic RNA segment (*Fodor et al., 1995*). The oligomeric viral nucleoprotein (NP) binds to the rest of the intramolecular loop of the viral genomic RNA to form a viral ribonucleoprotein complex (vRNP) (*Arranz et al., 2012*). The vRNPs are released into the host cytoplasm upon infection and migrate into the host nucleus for viral transcription (*Chou et al., 2013*; *Lakadamyali et al., 2003*). The vRNP initiates the viral transcription in the nucleus by deriving a 5′-capped short RNA fragment from the host nascent RNA on the stalled RNA polymerase II (Pol II) (*Walker and Fodor, 2019*). At the end of transcription, the viral polymerase adds a poly-A tail to the viral mRNA by stuttering on a short uridine repeat in the viral RNA template (*Poon et al., 1999*). Thus, eight viral segments sharing the common mechanism of viral transcription with no additional layers of transcriptional controls (e.g. binding of transcription factors), the transcription of eight influenza viral genes is likely to occur at the same rate across the segments that is determined by the collision of the vRNP and the stalled host Pol II.

Another possible factor that would determine the rate of transcription is the length of viral RNA template (*Enami et al., 1985*; *Phan et al., 2021*), given that the duration of transcription is rate-limiting. However, this does not agree with the order of protein expression in the influenza virus infection: PB2 (2.4 kb), PB1 (2.3 kb), and PA (2.2 kb), the three longest viral genes, and NP (1.8 kb) are expressed first, followed by the expression of three segments HA (1.8 kb), NA (1.5 kb), and M (1.0 kb) (*Lamb and Choppin, 1983*; *Hatada et al., 1989*).

A third possibility is the nuclear retention of viral mRNAs. Nuclear retention of mRNAs has gained an attention as a point of control on the protein expression in the context of transcription burst (*Bahar Halpern et al., 2015*). Previous studies reported the accumulation of M mRNAs in the nuclear speckles (*Mor et al., 2016*), NXF1-dependent nuclear export of HA, NA, and M mRNAs (*Read and Digard, 2010*; *Larsen et al., 2014*), and the nuclear export inhibition by the host factor hnRNPAB for M, HA, NP, and NS mRNAs (*Wang et al., 2021*). It is becoming clear that the influenza viral segments show differential dependencies on the nuclear export factors, and this dependency is not accounted for by the RNA structure (i.e. whether they are intron-less, unspliced, or spliced). Because an incomplete set of viral segments was studied in each of these studies, the influenza viral mRNA transport remains elusive.

In this paper, we introduce a statistical framework for systematically quantifying the nuclear transport across all the eight viral segments. We localised eight viral mRNAs simultaneously in single cells *in situ*. Intracellular distribution of viral mRNAs indicated that the nuclear export rate of viral transcripts varied according to the segments. A statistical model, exploiting cell-to-cell variation in the abundance of transcripts due to the stochastic nature of multiple virological processes prior to the onset of viral transcription, allowed for the estimation of nuclear export rate from the intracellular distribution of mRNAs observed at a single time point. With this model, we revealed the variation of the nuclear export rate in the eight viral segments. Notably, we show that HA and NA mRNAs were the slowest in exiting the nucleus, suggesting the possibility that influenza A virus uses nuclear retention of mRNAs to delay the production of antigenic molecules in the course of viral infection. This study demonstrates that the nuclear export rate of nascent transcripts can be estimated from a snapshot of mRNAs localised in single cells. Our finding using this model serves as a reference for the nuclear export rate of influenza viral transcripts across the segments, shedding light on the comprehensive understanding on the nuclear export of viral transcripts.

# Results

## Diffraction-limited, single-colour imaging of viral mRNAs

First, we used single-molecule RNA FISH (smFISH) to study the viral transcription of PB1, HA, NP, and NS. The lung carcinoma cell line A549 was infected with the WSN strain of influenza A virus at the multiplicity of infection (MOI) of ~2 for 2 hr, and the viral mRNAs were visualised with a tile of short oligonucleotide probes, each labelled with a single fluorophore (*Raj et al., 2008*; *Figure 1A*). Viral mRNAs were detected as single diffraction-limited spots in the three-dimensional (3D) image stacks. Each spot represents a single mRNA molecule (*Femino et al., 1998*), allowing for quantification of the absolute number of mRNA molecules in individual cells (*Figure 1B*).

We localised the mRNAs in 3D and counted the total number of mRNAs in each cell (*Figure 1C*). The median number of mRNAs per cell was 145, 231, 310, and 303 molecules for PB1 (2.3 kb), HA (1.8 kb), NP (1.8 kb), and NS (1.0 kb), respectively (*Figure 1D*), in line with the theoretical prediction that the rate of mRNA production is partly determined by the gene length.

What came intriguingly was the intracellular distribution of mRNAs of the four segments. The intracellular distribution of mRNAs projected on the XY plane (*Figure 1C*) indicated that the HA mRNAs were more abundant in the nucleus compared with the other three segments (*Figure 1E*).

We wished to quantify the nuclear ratio of mRNAs in each cell. The mRNA count within the 2D nuclear projection is considered to be a good approximation for the number of mRNAs contained in the nucleus (*Billman et al., 2017*; *Miura et al., 2019*). We validated this approximation for the lung epithelial A549 cells used in this study by looking at the intracellular Z positions of mRNAs: The mRNAs within the nuclear projection are expected to have higher Z positions over those of other mRNAs in the cell periphery, considering that the nucleus has the height of several microns, much thicker than the cytoplasm in the cell periphery. To this end, mRNAs were segmented into three regions in the 2D projection, i.e., nuclear, perinuclear, and peripheral, depending on the distance from the nucleus (*Figure 1F*). In contrast to the perinuclear (orange) and peripheral (green) mRNAs, mRNAs within the nuclear projection (blue) were distributed around the nuclear focal plane Z=0 (*Figure 1G*); about 80% of mRNAs in the nuclear projection (blue) were above the baseline (*Figure 1H*), defined by the median of the Z distribution of peripheral NP mRNAs (*Figure 1G*, red dashed line). Thus, in this paper, we consider the spots within the nuclear 2D projection to be the nuclear fraction for the sake of simplicity.

With this approximation, about 30% of the PB1, NP, and NS mRNAs were nuclear, while ~70% of HA mRNAs were in the nuclear fraction (*Figure 1I*), confirming the visual inspection that the HA mRNAs are more abundant in the nucleus at the early time point of infection (*Figure 1E*). This result indicates that the rate of viral mRNA export from the nucleus varies according to the viral segments. Note that our fluorescent probes could also detect complementary RNA (cRNA), an intermediate RNA that is produced for the viral genome replication, as it has the same polarity as viral mRNA. However, the fluorescent signals detected at 2 hr post-infection are deemed to be from the mRNAs because cRNA production occurs much later than the mRNA synthesis (*Phan et al., 2021*; *Kawakami et al., 2011*).

## Mapping eight viral mRNAs in single cells by multiplex RNA FISH

Next, we wish to simultaneously quantify all the eight viral segments *in situ*. To this end, we applied multiplex error-robust RNA FISH (MERFISH) (*Moffitt et al., 2016*), an imaging-based transcriptome assay in which multiple rounds of probe hybridisation and imaging are performed to identify hundreds to thousands of mRNA species *in situ* by decoding binary fluorescent signals. For example, hundreds of distinct mRNAs can be encoded in 16-bit binary signals, in which case eight rounds of hybridisation and imaging would be required to read out the signals with two fluorophores (*Moffitt et al., 2016*). In our case of detecting eight influenza A virus segments, 6 bits were sufficient to accommodate all the eight mRNAs, with each code being separated by two or more binary flips from others (*Figure 2A*). Thus, only three rounds of hybridisation and two-colour imaging were required. We sought for an economical way of performing sequential rounds of hybridisation and imaging, instead of setting up a custom-built, automated liquid pumping system used elsewhere (*Moffitt et al., 2016*). Inspired by an *in vitro* reconstitution assay of molecular motors (*Miura et al., 2010*), we assembled a flow cell on a slide glass, comprising two stripes of double-sided tape and the coverslip, with cells on the coverslip facing inwards (*Figure 2A*). Liquids were introduced from the right open side and wicked from the left. During image acquisition, the two open sides were sealed with rubber cement. This

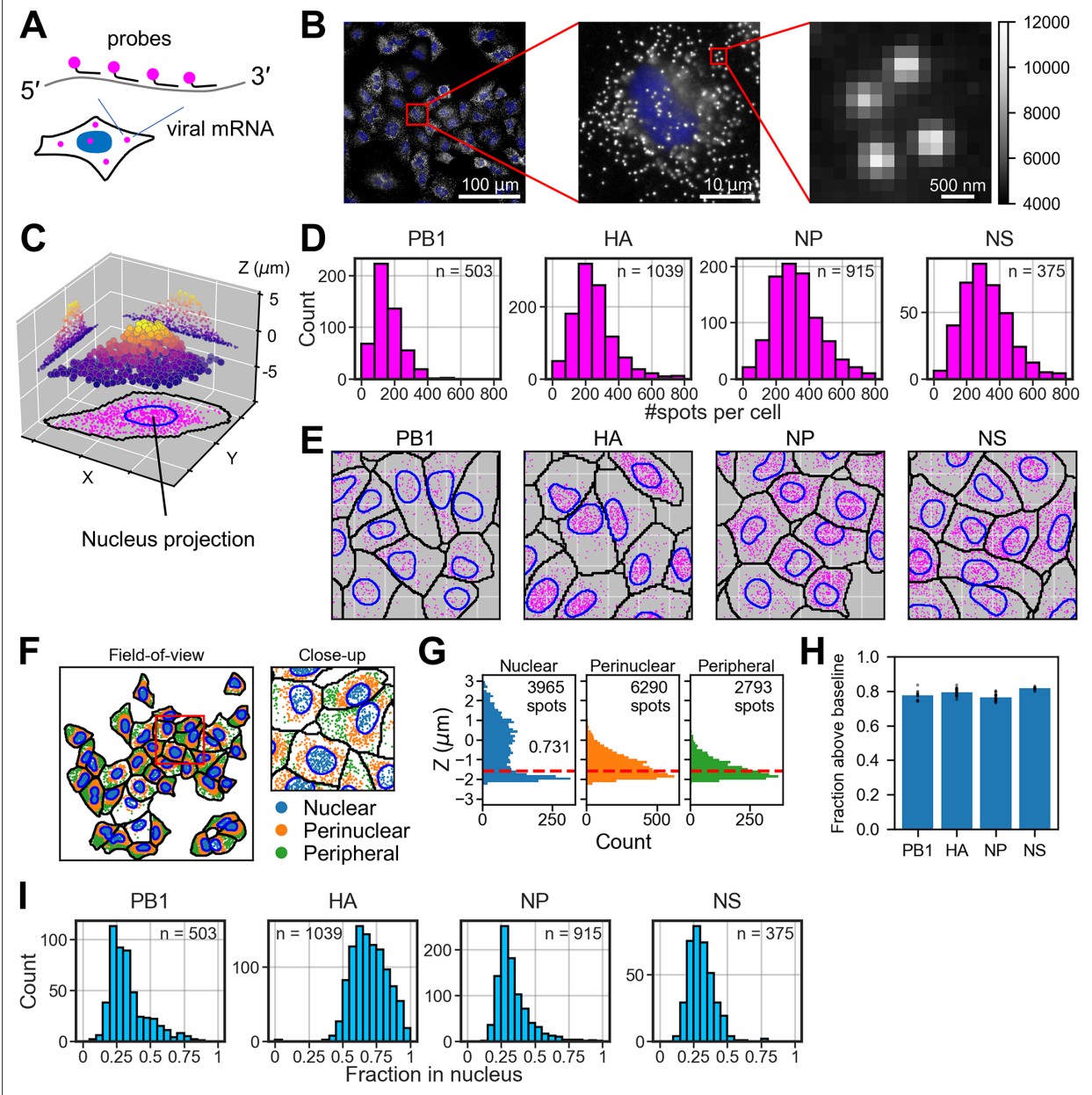

**Figure 1.** Viral mRNA detection by single-colour RNA FISH. (**A**) A schematic diagram of mRNA labelling for the diffraction-limited imaging. Each mRNA was hybridised with a tile of oligonucleotides, each labelled with a single fluorophore at the 3' end. (**B**) A representative field-of-view of the microscope image in which nucleoprotein (NP) mRNAs were stained with Quasar 670. White, single-molecule RNA FISH (smFISH) signal; blue, nucleus stained with DAPI. The red square indicates the area for the sequential close-up of the image. The signal intensity is shown on the same greyscale across the three panels, indicated on the right. (**C**) Spot detection in 3D. Spots are projected to either the XZ or YZ plane to indicate the distribution of spots in a typical 3D cell volume. The colour of each spot indicates the Z position of that spot. Spot projection to the XY plane at the bottom (magenta) highlights the XY positions of spots relative to the nucleus boundary (blue), i.e., nucleus projection on the XY plane. The grid interval on the X and Y axis is ca. 18.3 µm. (**D**) Total count of mRNAs in each cell. The sample size (n) indicates the number of cells segmented in the microscope images. (**E**) Intracellular distribution of viral mRNAs. Spots in the 3D volume were projected on the XY plane, together with the cytoplasmic boundary (black solid line) and the nucleus (blue solid line). The grid interval indicates ca. 18.3 µm. (**F**) Spot segmentation within cells. The panel on the left shows the entire field-of-view (measures ~294 µm), and the panel on the right shows the close-up of the area indicated in the red square (~73 µm). Spots within the 2D nuclear projection are indicated in blue (nuclear); spots within the expanded nuclear boundary twofold from the centre of mass of the nucleus in 2D are indicated in orange (perinuclear); and spots outside this expansion are indicated in green (peripheral). (**G**) Distribution of spots on the Z axis in the three segments (nuclear, perinuclear, and peripheral) presented in panel F. The red dashed line shows the median Z coordinates of the peripheral spots, used to indicate the baseline of spots. The number in the histogram for nuclear spots (0.731) indicates the fraction of spots that were above the baseline in the field-of-view presented in panel F. (**H**) Fraction of spots in the 2D nuclear segment that were above the baseline (the red dashed line in panel G, defined in the main

*Figure 1 continued on next page*

*Figure 1 continued*

text). Each dot indicates the fraction calculated in each field-of-view, and the blue bar indicates the mean. (**I**) Fraction of spots within the 2D boundary of the nucleus. The sample size is the same as in panel D.

---

easy implementation of flow cell allowed us to comfortably perform three rounds of hybridisation and two-colour imaging to read out the 6-bit binary tags; and by decoding the 6-bit encodings, the eight viral segments were identified in each cell (*Figure 2B*). On average, about 100–200 molecules of each segment were contained per cell (*Figure 2C*, top row). The difference in the total number of spots from the result presented in *Figure 1D* is likely due to the batch effect (experimental variation).

The nuclear fraction of mRNAs varied between the viral segments (*Figure 2C*, bottom row): While ~30% of mRNAs were nuclear for PB1, NP, and NS segments, about 70% of mRNAs remained in the nucleus for HA and NA segments; for the other segments (PB2, PA, and M), around half of mRNAs were in the nucleus. This reiterates the idea that the rate of viral mRNA export from the nucleus varies according to the viral segments. These results were confirmed by smFISH using probes for each segment that were synthesised economically in-house (*Figure 2—figure supplement 1A, B, and C*).

Single-cell clustering according to the abundance of eight viral segments revealed that the cell population was heterogeneous in terms of the abundance of viral mRNAs (*Figure 2D*). A pair-wise analysis of the abundance of viral segments indicated inequal distribution of viral mRNAs, regardless of the segments (*Figure 2E*): Cells that carry abundant mRNAs from one segment tend to also carry abundant mRNAs from the other segments. These observations suggest that the viral transcription is susceptible to the stochastic variation in each infection (*Heldt et al., 2015*; *Russell et al., 2018*), which we subsequently used in our statistical model to estimate the nuclear export rate.

## Conceiving the model

We suspected that the higher nuclear ratio of HA and NA mRNAs is due to the slower nuclear export rate for these mRNAs compared with other segments such as NP and NS. Thus, we wished to quantify the nuclear export rate of eight viral mRNA species from the observed nuclear-to-cytoplasmic distribution. To this end, we conceived a statistical model using the cell-to-cell variation in the abundance of viral transcripts of each segment (*Figure 3A*). In this model, influenza viral particles are added to the cell population at time 0 hr and are allowed to undergo viral attachment to the cell surface, membrane fusion, and vRNP migration to the nucleus. The waiting time for these processes vary in each infection due to the stochastic nature of biochemical events (*Lakadamyali et al., 2003*; *Heldt et al., 2015*; *Heldt et al., 2012*), and therefore, the onset of viral transcription varies from cell to cell in the population. In other words, each cell will have a varying duration for viral transcription by the time the observation is made at time 2 hr (*Figure 3A*, top). As a result, the population contains 'front runners' in which the viral transcription was initiated early, and therefore, abundant viral transcripts are carried; 'slow starters' in which the transcription began late, containing fewer transcripts; and many other cells that lie along the spectrum between the two. The cell population at the single point in time, therefore, forms a trajectory of the time development of viral transcription and subsequent nuclear export on the scale of nuclear fraction against the abundance of transcripts (*Figure 3A*, bottom): In a spontaneous transcription burst, the nascent transcripts emerge in the nucleus at the early stage of the burst, and they are exported to the cytoplasm as more transcripts are being synthesised (*Billman et al., 2017*).

Indeed, when the nuclear ratio of viral mRNA was plotted against the total quantity of mRNA molecules observed by multiplex RNA FISH in each cell, the plots formed a curve: It originated from top left (i.e. nearly 100% of the nuclear ratio when a cell contained a few mRNA molecules), and descended as the number of mRNA molecules increased (30–70%, depending on the segments) (*Figure 3B*, *Figure 3—figure supplement 1*).

We further validated this concept by performing smFISH measurements along the time-course with 40-min interval (*Figure 3C*). A549 was infected with WSN at the MOI of ~0.02. This low MOI allowed for spot quantification over longer time points without the spots becoming too crowded. At each time point, the NP mRNAs were stained and quantified. The total mRNA count was virtually zero at the beginning (0 and 40 min). As the mRNA count increased (80–200 min post-infection) (*Figure 3C*, top), the nuclear ratio against the total mRNA count obtained from the single cells progressed along a curve, from the top left to the bottom right (*Figure 3C*, bottom), supporting the concept of our statistical model.

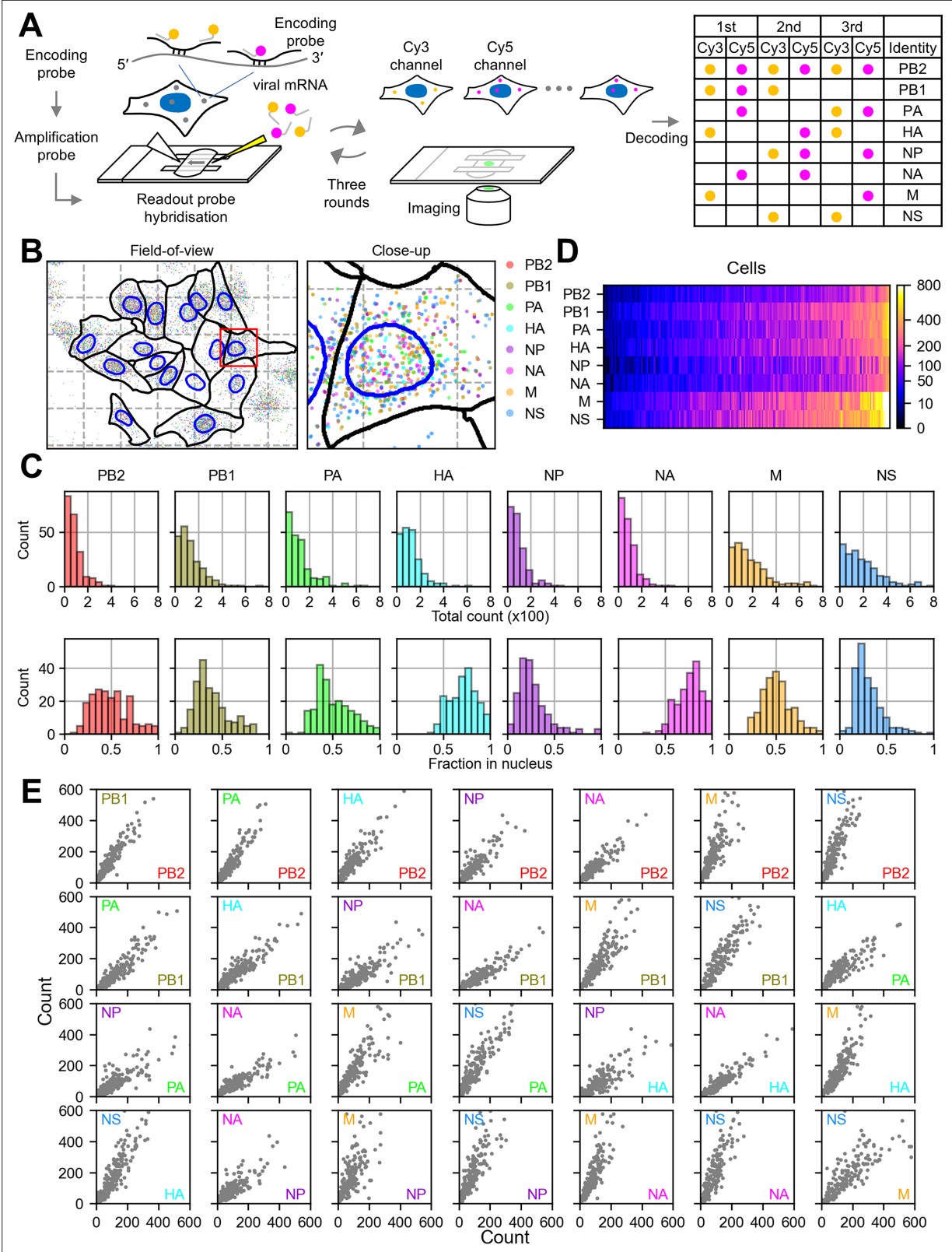

**Figure 2.** Viral mRNA detection using multiplex RNA FISH. (**A**) An overview of the assay. The viral mRNAs were hybridised with the encoding probes, and then the amplification probes were annealed (not shown in the diagram for simplicity). A flow cell was assembled in which the readout probes were hybridised and the cells were imaged in the three sequential rounds of hybridisation and imaging. Subsequently fluorescent signals were decoded according to the encodings listed on the right. (**B**) A representative mapping of identified viral mRNAs on the segmented cell and nuclear boundary in

*Figure 2 continued on next page*

*Figure 2 continued*

the entire field-of-view (left) and the close-up in the red square (right). The mesh grids indicate ~34 μm interval on the left, and ~17 μm interval on the right. Cells were removed if their cytoplasmic boundaries touched any edge of the field-of-view. (**C**) Total count of mRNAs in 204 single cells (top row) and the nuclear fraction (bottom row) for each segment. (**D**) Heatmap representation of mRNA counts in single cells. The viral segments are in rows, cells in columns. The cells were clustered column-wise to highlight the inequality in the abundance of transcripts carried per cell. The colour code for the absolute count is indicated in the vertical bar on the right. (**E**) A pair-wise analysis of the total transcript counts per cell between segments.

The online version of this article includes the following figure supplement(s) for figure 2:

**Figure supplement 1.** Independent validation of the total mRNA count and the nuclear ratio of each segment by single-molecule RNA FISH (smFISH).

## Implementing a kinetic model to estimate the nuclear export rate

The curvature of the trace varied between segments (***Figure 3B***). We explored to see whether we could extract the nuclear export rate for each segment from the trace formed by the observed data. To this end, we formulated a simple kinetic model in which mRNAs are synthesised at the rate $\lambda$ in the nucleus, and are exported with the reaction constant μ (***Figure 4A***). In this model, the molecules of mRNAs in the nuclear and cytoplasmic compartments are described, respectively,

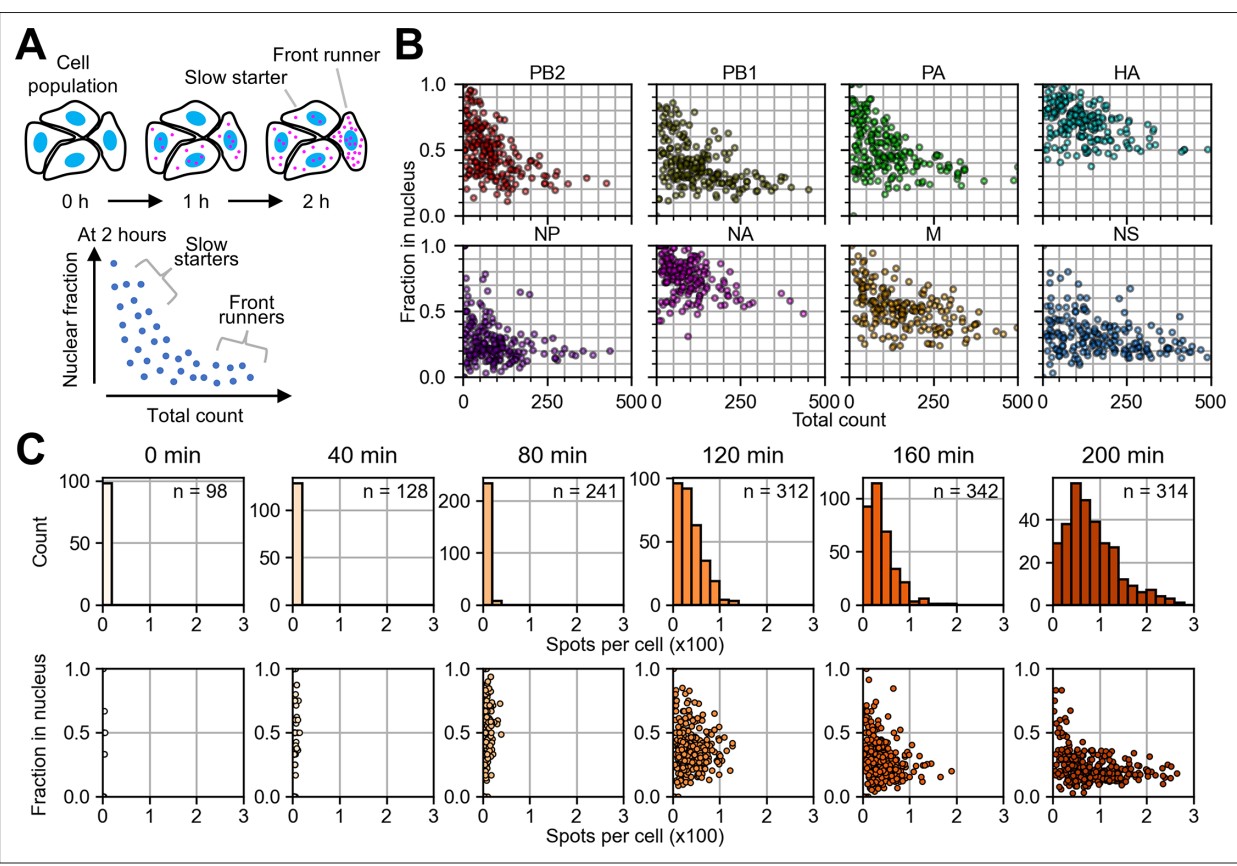

**Figure 3.** Cell-to-cell heterogeneity for time-course inference. (**A**) A theoretical diagram of how viral mRNAs emerge in single cells in the population. The front runners designate ones in which the viral transcription was initiated earlier than others, and the slow starters are ones that started transcription lately. The bottom plot shows the theoretical relationship between the nuclear ratio and the abundance of transcripts in which the front runners and slow starters (and many more along the spectrum in between) compose the clusters as indicated. (**B**) Scatter plots of nuclear fraction against the total count in each cell, derived from the dataset presented in ***Figure 2C***. (**C**) Time-course of the distribution of total nucleoprotein (NP) mRNA count per cell, measured by single-molecule RNA FISH (smFISH), at the indicated time points (top row); and the change in the nuclear fraction size, plotted against the total number of transcripts (bottom row). The sample size (n) indicates the number of cells observed at each time point.

The online version of this article includes the following figure supplement(s) for figure 3:

**Figure supplement 1.** Independent validation of the nuclear ratio against the total count of mRNAs for each segment by single-molecule RNA FISH (smFISH).

**eLife** Research article

Microbiology and Infectious Disease

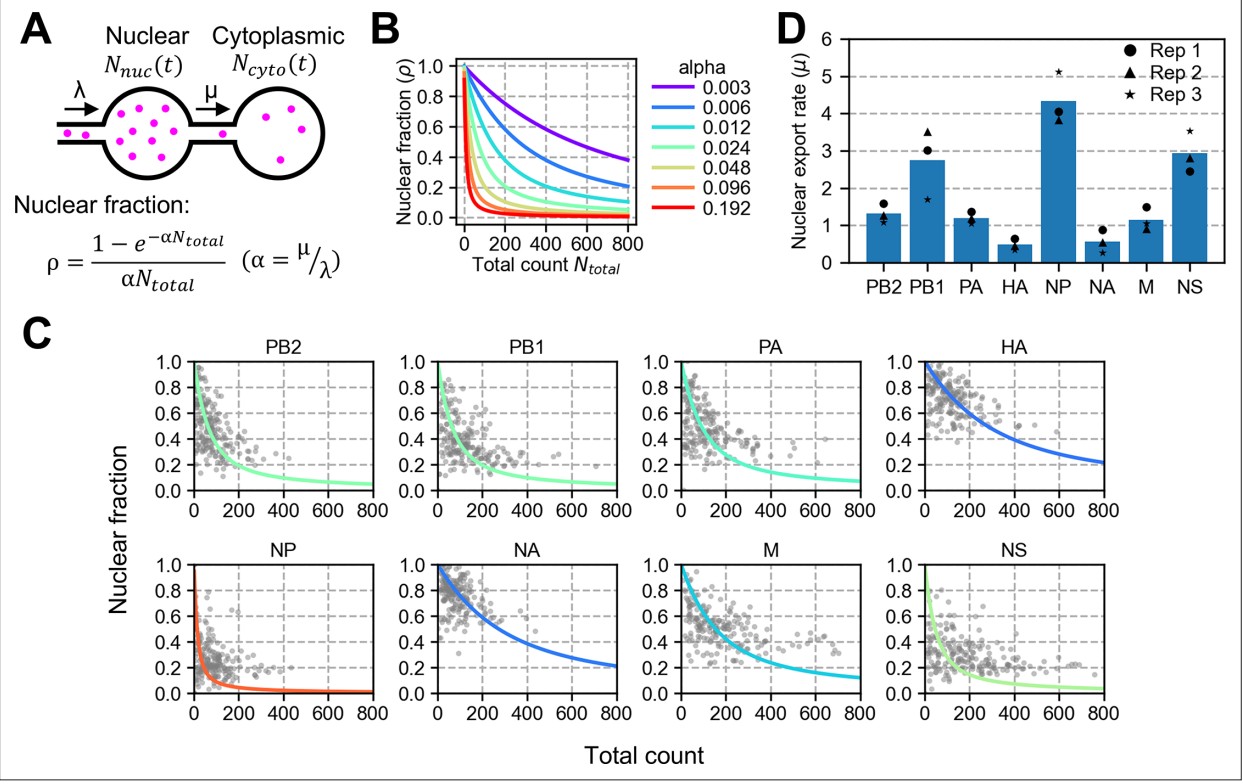

**Figure 4.** Kinetic model for estimating the nuclear export rate. (**A**) A schematic diagram representing the production of viral transcripts in the nucleus and the export to the cytoplasm. The two spherical compartments are for the nucleus and the cytoplasm. Viral transcripts (magenta) are produced at a rate constant $\lambda$ in the nucleus, and exported at a rate constant $\mu$ to the cytoplasm. (**B**) The relationship between the nuclear fraction and the total count in the kinetic model, according to the formula presented in panel A. The parameter $\alpha$ determines the curvature of the plot. (**C**) The parameter fit to the observed data points. The parameter $\alpha$ was estimated by fitting the equation to the observed data points presented in *Figure 3B*. The colour representation in each line is the same as the ones used in panel B, according to the parameter $\alpha$. (**D**) Estimated nuclear export rate. The parameter $\mu$ was estimated from the three independent assays (indicated in black for each segment); the blue bar indicates the mean of the three measurements.

$$\frac{d\left[Nuclear\right]}{dt} = \lambda - \mu\left[Nuclear\right] \tag{1}$$

$$\frac{d\left[Cytoplasmic\right]}{dt} = \mu\left[Nuclear\right] \tag{2}$$

Omitting the degradation of mRNAs (see the next section), the analytical solutions of these equations were as follows:

$$\left[Nuclear\right] = \frac{\lambda}{\mu}\left(1 - e^{-\mu t}\right) \tag{3}$$

$$\left[Cytoplasmic\right] = \lambda t - \frac{\lambda}{\mu}\left(1 - e^{-\mu t}\right) \tag{4}$$

Thus, the mRNA nuclear fraction $\rho$ was described as a function of total mRNA molecules $N_{total}$,

$$Nuclear\,fraction\,\rho = \frac{1}{\alpha N_{total}}\left(1 - e^{-\alpha N_{total}}\right) \tag{5}$$

where

$$\alpha = \frac{\mu}{\lambda} \tag{6}$$

The nuclear fraction $\rho$ described in *Equation 5* is time-independent, and the parameter $\alpha$ determines the curvature of the line that the single cells would trace in the time-course (*Figure 4B*). Thus, we estimated the parameter $\alpha$ by fitting to the observed data points derived at the time in which cells

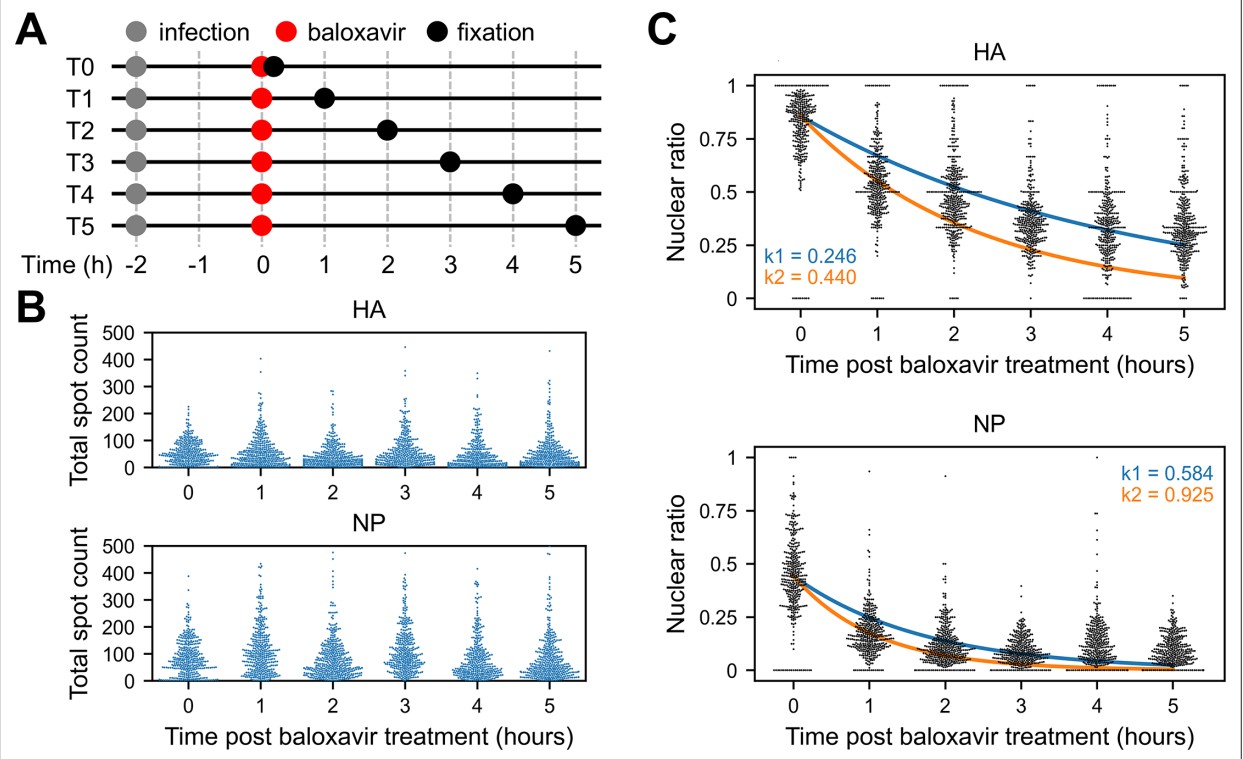

**Figure 5.** Time-course experiment for viral mRNAs. (**A**) A schematic diagram of the time-course measurement. Cells were infected with the virus 2 hr prior to the baloxavir treatment for halting the viral transcription. Cells were fixed at 1-hr interval thereafter, then subjected to single-molecule RNA FISH. (**B**) The total mRNA counts over time post baloxavir treatment for the hemagglutinin (HA) (top) and nucleoprotein (NP) (bottom) segments. (**C**) The change of mRNA fraction detected in the nucleus over time post baloxavir treatment. The exponential decay was fitted against the data using either all the data points (indicated with the blue line) or the first two time points (indicated with the orange line). The exponents (i.e. the nuclear export rates) estimated are shown on the panel with the corresponding colour.

The online version of this article includes the following figure supplement(s) for figure 5:

**Figure supplement 1.** Time-course measurement of viral mRNAs.

**Figure supplement 2.** Viral mRNA export.

were fixed (**Figure 4C**). By obtaining the synthesis rate $\lambda$ from the total number of mRNAs produced by the time of fixation (**Figure 2C**, top row), the parameter μ for each segment was obtained according to **Equation 6** (**Figure 4D**). The parameter μ represents the fraction of mRNAs in the nucleus that are exported within an infinitesimal time (i.e. the nuclear export rate). The result reveals the variation of nuclear export rate in the eight viral segments (**Figure 4D**). The nuclear export rate μ agreed with the historical classification of the early and late influenza A virus genes (**Lamb and Choppin, 1983**; **Hatada et al., 1989**), suggesting the possibility that the timing of influenza A virus protein expression in the initial stage of infection is determined by the rate of the nuclear export.

## Time-course study of viral mRNA degradation and nuclear export

We omitted the mRNA degradation in the kinetic model described in the previous section. To confirm that this simplification can be justified, we measured the viral mRNAs over 5 hr after halting the viral transcription (**Figure 5A**). The influenza virus was allowed for transcription for 2 hr, after which the transcription was blocked by baloxavir (**Noshi et al., 2018**), inhibiting the nuclease activity that resides in the viral polymerase subunit PA. We examined the HA and NP segments by smFISH, the two extreme examples that showed the highest and lowest nuclear export rate in our model, and found no reduction in viral mRNAs over the 5 hr after inhibiting the viral transcription (**Figure 5B**, **Figure 5—figure supplement 1**).

Next, we estimated the mRNA export rate using the time-course measurement. The nuclear ratio of viral mRNAs was plotted at each time point, and the decay constant of the exponential decay

(i.e. the rate of nuclear export, corresponding to the parameter μ in our model) was fitted against the nuclear ratio that declined over time (*Figure 5C*, *Figure 5—figure supplement 2*). For the HA segment, the nuclear export rate estimated using all the time points (indicated with the blue line) was 0.243 hr⁻¹ (the mean of three replicates), in a good agreement with the one estimated in our model. On the other hand, for the NP segment, the nuclear export rate estimated from the time-course measurement was 0.631 hr⁻¹ (the mean of three replicates), which was much lower compared with our model estimation for the NP segment. We gave the reason that the NP mRNAs were exported so rapidly that, in the time-course study, the nuclear ratio had already started to approach its lowest by the first time point following the baloxavir treatment, leading to the underestimation of the mRNA export rate. Using only the first two time points for the decay-constant fit (indicated with the orange line) did not make a significant difference (estimated rate 0.972 hr⁻¹ for the NP segment, the mean of three replicates). This highlights that our model enables the estimation of mRNA export rate from a single snapshot of mRNA distribution in cells, eliminating the need for time-course measurements that could potentially underestimate the export rate.

## Discussion

Utilising the stochastic nature of influenza viral transcription (*Heldt et al., 2015*; *Russell et al., 2018*), we introduced a quantitative framework for estimating the nuclear export rate of viral mRNAs to advance our current understanding on the kinetics of viral gene expression. This was achieved by *in situ* localisation of eight influenza A virus segments simultaneously, coupled with the statistical model that allowed for estimating the nuclear export rate from the intracellular distribution of mRNAs at a single time point. This is based on the concept that the cells in the population are at various stages of transcription at the time the observation is made, due to the multiple stochastic processes the virus needs to overcome before it begins transcription (*Lakadamyali et al., 2003*). We propose that the initiation of each viral transcription is concurrent in the eight viral segments; it is instead the nuclear export rate that controls the temporal protein expression at the initial phase of infection. The early production of the polymerase subunits (PB2, PB1, and PA) and NP would facilitate the formation of vRNP, and hence the viral transcription and genome replication. On the other hand, pushing back the peak of protein expression of viral antigens HA and NA on the cellular surface might be beneficial for the virus to delay the Fcγ-receptor-mediated, antibody-dependent cellular cytotoxicity (*DiLillo et al., 2016*; *Ko et al., 2021*) against the infected cells in which the virus is being replicated.

Instead of time-series measurements, the estimation of mRNA export rate was made from a snapshot of *in situ* RNA localisation data taken at a single time point. We anticipate that the statistical framework for quantifying the nuclear export rate proposed in our study has broad applications beyond the influenza virus infection. This benefits the research area on the regulation of gene expression, especially since nuclear retention has been recognised as a point of control against the transcription burst (*Bahar Halpern et al., 2015*). For example, it is applicable for the cell population in which *de novo* synthesis of mRNAs occurs concurrently, such as a stimuli-induced activation of immediate-early genes (*Miura et al., 2019*; *Kulkarni et al., 2018*). In our use case for the influenza virus infection, synchronised infection was not necessary, such as incubating cells with the virus at 4°C to facilitate attachment and subsequently shifting to 37°C to allow viral entry. The inherent heterogeneity in vRNP migration to the nucleus still remains (*Chou et al., 2013*; *Lakadamyali et al., 2003*). This randomness does not compromise our model; rather, our model exploits this random arrival of each vRNP in each cell in the system. This variation, in turn, generates cells carrying varying amounts of transcripts, enabling the estimation of nuclear export rate. Importantly, more variation ensures the broader distribution of transcript levels, allowing for more precise parameter fitting in our model. It is also important to note that our model does not require the correlation between segments. Our model estimates the export rate of each mRNA species individually. On the contrary, this approach is not appropriate for scenarios in which transcription bursts occur repeatedly and randomly in each individual cell within the population (*Billman et al., 2017*; *Kiik et al., 2022*); cells at the late stage of transcription burst, having residual amount of transcripts in the cytoplasm (*Billman et al., 2017*), would obscure the time-course trajectory of the nuclear export.

The experiment in this study was conducted at the early stage of infection (up to 200 min post-infection). This conferred two advantages over investigating influenza viral mRNA export at the late stage of infection (e.g. 6–12 hr post-infection). First, our study allowed for the absolute quantification

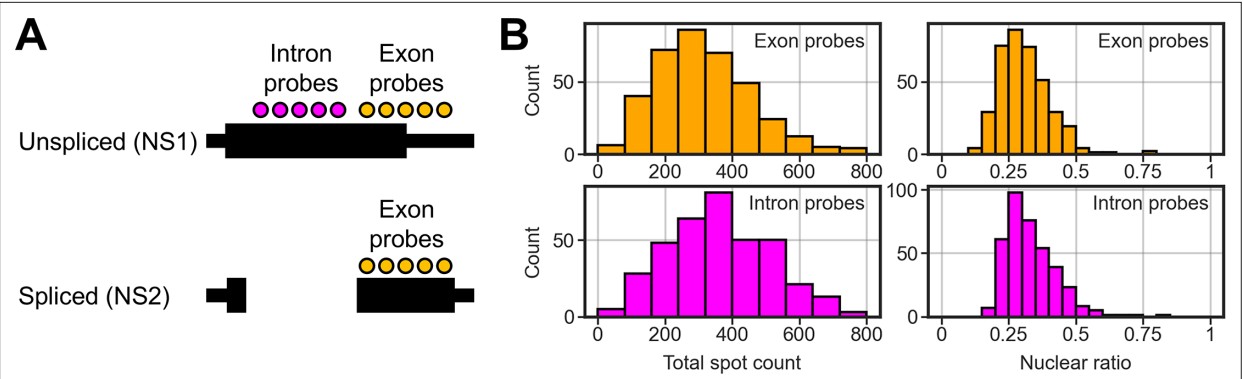

**Figure 6.** Exclusive detection of unspliced NS transcripts. (**A**) Probe-binding sites on the NS segment. The exon probes, used in *Figure 1*, target the second exon of NS2 (spliced form), and the intron probes are against the region that is spliced out. Thus, exon probes detect both NS1 and NS2, while intron probes exclusively detect NS1. The black line represents the NS gene. The protein coding region is indicated with the wide horizontal bar. (**B**) The total spot count (left column) detected using either exon probes or intron probes. The spot nuclear ratio in each cell is indicated on the right. Note that the plots of the exon probes (orange) are the reproduction from *Figure 1* and are presented for comparison with those of the intron probes (magenta).

of viral transcripts at the single-molecule level in each cell and, thereby, enabled us to formulate the model and fit the equation to the numerical data points. Otherwise at the late stage of infection, accurate quantification of mRNAs by the imaging-based single-molecule detection would not be feasible due to the high density of mRNAs. Second, at the late stage of infection, NS1 protein would be expressed in the cytoplasm. NS1 protein would then migrate into the nucleus and alter the nuclear export of viral mRNAs (*Pereira et al., 2017*), obscuring the inherent nature of nuclear export of each viral segment. Thus, minimising the impact of NS1 protein in our study, we elucidated the quantitative measure for the influenza viral mRNA nuclear export that is intrinsic to each viral segment.

Host factors that relay the influenza viral mRNAs to the nuclear pore remain elusive. NXF1 and CRM are two major cellular mRNA export pathways for the 5′-capped mRNAs (*Palazzo and Lee, 2018*). Of these two pathways, influenza A virus mRNAs, also carrying 5′-cap structure derived from the host nascent mRNA and the poly-A tail as a result of viral polymerase stuttering the uridine repeat, are dependent on the NXF1 pathway (*Read and Digard, 2010*; *Larsen et al., 2014*). However, the eight viral segments show differential dependencies on this export pathway. The segments HA, M, and NS are exported by NXF1 and UAP56, whereas PB2, PB1, and PA appear to be independent of the NXF1 pathway (*Larsen et al., 2014*). The nuclear export rate derived in our study does not agree with the dependencies on NXF1. According to the previous study, both HA and NS are dependent on NXF1 and UAP56 (*Read and Digard, 2010*); however, in our study, HA and NS had significantly different nuclear export rate.

Two viral segments M and NS are subject to splicing, yielding unspliced and spliced transcripts. Our current system lacks the ability of discriminating between the unspliced and spliced forms of these segments; although we infer that the majority of NS segments that we detected in this study were in the unspliced form because we obtained similar result with probes exclusively detecting the unspliced form of NS (*Figure 6*). An observation that the unspliced form of M and NS dominates throughout the infection supports this (*Phan et al., 2021*). In either case, the RNA structure (whether spliced or unspliced) does not appear to accord with the NXF1 dependency, nor the nuclear export rate, either.

Recently, the human nuclear factor hnRNPAB was shown to associate with the NP of influenza A virus and inhibit the NXF1 pathway. It is tempting to speculate that the host factors (e.g. hnRNPAB) have differential binding affinities to each segment, thus determining the rate of nuclear export (or the degree of nuclear export inhibition). It remains currently unknown what factors bind to the early genes such as PB2, PB1, and PA. Further studies will be required to identify the host factors that bind to these transcripts for the nuclear export.

We suggest that the nuclear export of influenza viral transcripts is intrinsic to the nucleotide sequence of each segment. A recent study revealed the internal sequence in the influenza virus HA mRNA that determined its susceptibility to the nuclear export via TREX-2 at the nuclear pore complex (*Bhat et al., 2023*). This finding offers promising insights into the intrinsic sequences encoded in influenza viral transcripts. Our study serves as a reference for the nuclear export rate of each segment

and, thus, sheds light to comprehensive understanding on the nuclear export of viral segments. By revealing the nuclear retention of influenza viral transcripts at the unprecedented resolution, this study opens up a new research direction on how the nuclear retention is regulated across the segments and its implication in viral pathogenesis.

# Materials and methods

**Key resources table**

| Reagent type (species) or resource | Designation | Source or reference | Identifiers | Additional information |
|---|---|---|---|---|
| Strain, strain background (influenza virus) | WSN | PMID:10430945 | Influenza A virus (A/WSN/1933(H1N1)) | |
| Cell line (*Homo sapiens*) | A549 | ATCC | CCL-185 | |
| Commercial assay or kit | Wash Buffer A | Stellaris | SMF-WA1-60 | |
| Commercial assay or kit | Wash Buffer B | Stellaris | SMF-WB1-20 | |
| Commercial assay or kit | Hybridization Buffer | Stellaris | SMF-HB1-10 | |
| Chemical compound, drug | Baloxavir acid | Shionogi | SH-02 | |
| Other | Coverslip | Matsunami Glass | C0180HT | 0.17±0.005 mm thickness |

## Cells and virus preparation

The human lung carcinoma cell line A549 was maintained in Dulbecco's Modified Eagle's Medium (DMEM) (Sigma, D5796) supplemented with 10% FBS (Nichirei Biosciences, 175012), 100 U/ml penicillin, and 100 µg/ml streptomycin (Gibco, 15140-122) at 37°C in 5% $CO_2$. Madin-Darby canine kidney (MDCK) cell line was maintained in Minimum Essential Medium (MEM) (Sigma, M4655) supplemented with 10% FBS, 100 U/ml penicillin, and 100 µg/ml streptomycin at 37°C in 5% $CO_2$. Influenza A virus strain WSN was propagated in MDCK cells in a 75 cm² flask. The plaque-forming unit of the harvested virus was determined using MDCK cells grown confluent in a six-well plate.

## Coverslip preparation

An 18-mm round coverslip (0.17±0.005 mm thickness, Matsunami Glass) was used. For MERFISH, two straight sides were created by cutting two edges off the glass using a laser-wheel glass cutter (Nikken Dia, NC-X03 LASER 110°). The coverslip was rinsed in 70% ethanol and placed in a 12-well plate. A549 cells were seeded and attached onto the coverslip overnight. On the following day, cells were rinsed twice with PBS and infected with WSN at the indicated MOI in MEM for 2 hr (unless indicated otherwise in *Figure 3C*) in the absence of FBS. After the infection, cells were fixed with 4% paraformaldehyde (Electron Microscopy Sciences, 15714-S) in PBS for 10 min and then rinsed three times with PBS. The coverslip was immersed in 70–75% ethanol and stored at −20°C until use.

For the baloxavir treatment in *Figure 5*, cells were infected with WSN for 2 hr as described above. Then, the medium with the virus was removed, and 800 µl of DMEM (containing 10% FBS, 100 U/ ml penicillin, and 100 µg/ml streptomycin) with 100 nM baloxavir acid (Shionogi, SH-02) was added to each well. Subsequently, coverslips were prepared by fixing the cells at the intervals indicated in *Figure 5A*.

## smFISH

The probes used in *Figure 1*, *Figure 3C*, *Figure 2—figure supplement 1* (probes for NP), and *Figure 6* were obtained from LGC Biosearch Technologies. Stellaris RNA FISH Probe Designer (https://www.biosearchtech.com/stellaris-designer) was used to design the probes against the coding region of WSN mRNAs.

The coverslip was equilibrated in 1 ml of Wash Buffer A (Stellaris) for >5 min in a 12-well plate prior to hybridisation. Cells were hybridised to 125 nM probes in Hybridization Buffer (Stellaris) on a Hybrislip (Sigma, GBL712222) at 37°C overnight in a humidified chamber. On the following day, the coverslip was rinsed twice in 1 ml of Wash Buffer A (Stellaris), each for 30 min. Cytoplasmic stain (HCS CellMask Green Stain, Invitrogen, H32714) was included in the second wash at the concentration of 50 ng/ml in *Figures 1, 3C, and 6*. Then, the coverslip was rinsed in 1 ml of Wash Buffer B (Stellaris) for

5 min. The coverslip was mounted in anti-fade medium containing DAPI (Vector Laboratories, H-1200) on a slide glass and sealed with nail polish.

For an economical reason and to allow for more flexible design of probes to increase the number of fluorophores attached per mRNA molecule, the probes used in *Figure 2—figure supplement 1* (except the NP probes) and *Figure 5* were synthesised in-house (*Figure 2—figure supplement 1A*). (The NP probe used in *Figure 2—figure supplement 1* was purchased from LGC Biosearch.) Candidate probe-binding sites in the coding region of each viral segment were identified using OligoMiner (*Beliveau et al., 2018*) and were further selected against the human RefSeq. The sequences were flanked by P5 (5′ AAT GAT ACG GCG ACC ACC GA 3′) and P7 (5′ CAA GCA GAA GAC GGC ATA CGA GAT 3′), with P7 at the 5′ end and P5 at the 3′ end of the probe sequence (*Figure 2—figure supplement 1A*). For the probes against HA, NA, M, and NS, additional 20-nt sequences RS0332 (5′ GGG AGA ATG AGG TGT AAT GT 3′) and RS0406 (5′ GAT GAT GTA GTA GTA AGG GT 3′) (*Moffitt et al., 2016*) were inserted between P7 and the probe, the probe and P5, respectively (*Figure 2—figure supplement 1A*). These two insertions were used for additional probes to bind to increase the number of fluorophores per molecule. An oligonucleotide pool comprising the set of probe-binding sites with these additional nucleotide sequences was synthesised at Integrated DNA Technologies (IDT). The oligonucleotide pool was subjected to PCR amplification (98°C for 30 s; 18 cycles of 98°C for 5 s, 64°C for 10 s and 72°C for 20 s; and 72°C for 5 min) (Phusion High-Fidelity DNA Polymerase, NEB, M0530S) using the P7 and a T7-fused P5 primer (5′ TAA TAC GAC TCA CTA TAG GGA ATG ATA CGG CGA CCA CCG A 3′) (*Figure 2—figure supplement 1A*). RNA was synthesised using T7 RNA polymerase (HiScribe T7 High Yield RNA Synthesis Kit, NEB, E2040S). Then, RNA was reverse-transcribed using a recombinant Moloney leukemia virus reverse transcriptase (GeneAce cDNA Synthesis Kit, Nippon Gene, 319-08881) with an ATTO550-conjugated P7 primer at the 5′ end. The single-stranded DNA probe was purified using a spin column (Zymo Research, D7010) after RNA digestion with a cocktail of RNase H and RNase A. The probe was further purified by Urea-PAGE to remove any residual RNA and the fluorescent-labelled primer.

The coverslip was equilibrated in 1 ml of 30% formamide buffer composed of 2×SSC and 30% deionised formamide (Sigma, F9037) in a 12-well plate for >5 min prior to hybridisation. Cells were hybridised with the probes at the concentration of 5.6 nM per probe in 100 µl Hybridisation buffer composed of 2×SSC, 30% deionised formamide, 0.1% yeast tRNA (Invitrogen, 15401-011), 1% (vol/vol) RNase inhibitor (NEB, M0314S), and 10% (wt/vol) dextran sulfate (Merck, S4030) (*Moffitt et al., 2016*) on a Hybrislip at 37°C overnight in a humidified chamber. On the following day, the coverslip was rinsed twice in 1 ml of 30% formamide buffer, each for 30 min. DAPI was included at 5 ng/ml in the second wash if additional staining was not performed with the external probes (for PB2, PB1, and PA). In this case, the coverslip was rinsed in 1 ml of PBS after the DAPI staining, mounted in anti-fade medium (Vector Laboratories, H-1900) on a slide glass, and sealed with nail polish. For staining HA, NA, M, and NS with the extra probes, the coverslip was first equilibrated with 10% formamide buffer composed of 2×SSC, 10% deionised formamide, and 0.05% (vol/vol) RNase inhibitor, and then hybridised with 10 nM each of probes for RS0332 and RS0406 (*Moffitt et al., 2016*), both labelled with Quasar 570 (LGC Biosearch) at the 3′ end, in Stellaris hybridisation buffer (containing 10% formamide) at 37°C in the humidified chamber for 1 hr. Cells were washed in 10% formamide buffer twice, with DAPI included in the second wash. The coverslip was rinsed in PBS, mounted in the anti-fade medium (Vector Laboratories, H-1900), and sealed with nail polish.

## Sequential hybridisation and imaging in the flow cell

An oligonucleotide pool for synthesising the encoding probes used in *Figure 2*, comprising the target-binding sequence, readout tags compatible with the two-step amplification probes (*Moffitt et al., 2016*; *Xia et al., 2019*), and the P5 and P7 PCR tags, were obtained from IDT. The target-binding sites were identified in the coding region of each segment using OligoMiner (*Beliveau et al., 2018*) and selected against human RefSeq. The readout tags RS0332, RS0343, RS0406, RS0255, RS0015, and RS0384 (*Moffitt et al., 2016*; *Xia et al., 2019*) were inserted adjacent to the upstream or downstream of the target-binding sites for the 6-bit encoding of each segment (*Figure 2A*). Probes were synthesised by PCR, T7 transcription, and reverse transcription as described above.

A coverslip was equilibrated in 1 ml of 30% formamide buffer (2×SSC and 30% deionised formamide) (*Moffitt et al., 2016*). Cells were hybridised with 1.6 µM encoding probes in Hybridisation

buffer composed of 2×SSC, 30% deionised formamide, 0.1% yeast tRNA, 1% RNase inhibitor, 10% dextran sulfate (*Moffitt et al., 2016*) on a Hybrislip at 37°C overnight in a humidified chamber. Cells were washed twice in 1 ml of 30% formamide buffer at 37°C, and then equilibrated in 1 ml of Readout-probe hybridisation and wash buffer composed of 2×SSC, 10% deionised formamide, and 20 units/ml Murine RNase inhibitor (*Moffitt et al., 2016*). The first and second amplification probes for RS0332, RS0255, RS0343, RS0406, RS0015, and RS0384, described previously (*Xia et al., 2019*), were annealed sequentially at 10 nM of each probe in Readout-probe hybridisation and wash buffer for 2 hr at 37°C on a Hybrislip. The coverslip was washed for 30 min twice in Readout-probe hybridisation and wash buffer at each of the first and second probe hybridisation. At the final wash, 0.10 µm green-fluorescent carboxyl polystyrene beads (Bangs Laboratories, FCDG002) were included (~0.1% vol/vol) as a fiducial marker for image registration. The cells and beads were fixed with 3.2% paraformaldehyde in PBS for 10 min. The coverslip was rinsed in PBS three times and stored in 2×SSC supplemented with 40 units/ml RNase inhibitor at 4°C overnight.

On the following day, a flow cell was assembled on a slide glass, composed of two parallel stripes of double-sided tape (0.23 mm thickness, ~5 mm apart) and the coverslip on top, with cells facing inwards (*Figure 2A*). The cells at the top and bottom end of the coverslip were scraped using a cell scraper before the assembly and any trace amount of liquid remaining on the exposed surface was removed with filter paper to ensure that the coverslip is firmly attached to the tape. The chamber was immediately flushed with 2×SSC to avoid cells drying up.

Readout-probe hybridisation and wash buffer was flushed in the chamber to equilibrate the cells. Readout probes were introduced at 10 nM in Readout-probe hybridisation and wash buffer and incubated at 37°C in a humidified chamber for 30 min. Subsequently, Readout-probe hybridisation and wash buffer was flushed to wash away the probes. Imaging buffer, composed of 2×SSC, 50 mM Tris-HCl pH 8.0, 10% glucose (Fujifilm Wako, 049-31165), 2 mM Trolox (Sigma, 238813), 0.5 mg/ml glucose oxidase (Sigma, G2133), 40 µg/ml catalase (Sigma, C30), and 0.1% RNase inhibitor (*Moffitt et al., 2016*), was flushed to the flow cell prior to the image acquisition. The flow cell was sealed at the both sides with Fixogum rubber cement ejected through a 200-µl pipette tip during the image acquisition.

After the imaging, the disulfide bond linking the probe and the fluorophore was cleaved with Cleavage buffer comprising 2×SSC and 50 mM TCEP (Sigma, 646547) for 15 min. The chamber was flushed with Readout-probe hybridisation and wash buffer to wash away the free fluorophores prior to introducing the readout probes for the next round. Readout probes used in this study are listed in *Table 1*.

## Imaging platform

Images were acquired using a Nikon Ti2-E with a Nikon Plan Apo Lambda ×60 (NA 1.40) (*Figures 1 and 6*) or ×100 (NA 1.45) (*Figure 3C*) objective lens equipped with a Prime 95B 25MM back-illuminated sCMOS camera (Photometrics). Otherwise, an Olympus IX83 equipped with an Olympus UPlan XApo ×60 (NA 1.42) objective lens and a cooled monochrome CCD camera (DP80, Olympus) was used (*Figure 2*, *Figure 2—figure supplement 1*, *Figure 3—figure supplement 1*, *Figure 5*, *Figure 5—figure supplement 1*, and *Figure 5—figure supplement 2*).

**Table 1.** Readout probes used in this study.

| ID | Round | Channel | Sequence (5' to 3') and fluorescent dye | Manufacturer |
|---|---|---|---|---|
| RS0332amp | 1 | 1 | TTC TTC CCT CAA TCT TCA TC -S-S- Q570 | LGC Biosearch |
| RS0343amp | 1 | 2 | TAC TAC AAA CCC ATA ATC CC -S-S- Q670 | LGC Biosearch |
| RS0406amp | 2 | 1 | AAT CTC ACC TTC CAC TTC AC -S-S- Q570 | LGC Biosearch |
| RS0255amp | 2 | 2 | TCA CCT CTA ACT CAT TAC CT -S-S- Q670 | LGC Biosearch |
| RS0015amp | 3 | 1 | TCT CAC ACC ACT TTC CTC AT -S-S- ATTO550 | Generay |
| RS0384amp | 3 | 2 | TCC TCA TCT TAC TCC CTC TA -S-S- ATTO647N | Generay |

## Spot quantification

Fluorescent spots detected by smFISH were quantified using either FISH-QUANT (*Mueller et al., 2013*; *Figures 1, 3C, and 6*) or a more recent Python package Big-FISH (*Imbert et al., 2022*; *Figure 2—figure supplement 1*, *Figure 3—figure supplement 1*, *Figure 5*, *Figure 5—figure supplement 1*, and *Figure 5—figure supplement 2*). Cytoplasmic and nuclear segmentation was performed using Cellpose (*Stringer et al., 2021*; *Pachitariu and Stringer, 2022*). Objects that were on the boundary of the image was eliminated. Cytoplasmic regions with no nucleus therein due to the erroneous segmentation were also eliminated. The identification numbers assigned to each object by Cellpose were rearranged so that the cytoplasm and nucleus of the same cell correspond. In order to make the Cellpose mask readable in FISH-QUANT, the Cellpose mask were converted into X and Y coordinates by tracing each object in the mask using the Moore neighbourhood edge-finding algorithm.

For MERFISH signal decoding, signals from the green-fluorescent carboxyl beads were used to register the images. Images at the corresponding field-of-view from the three rounds of imaging were registered using phase_cross_correlation from the Python package skimage.registration. The registered images were filtered using Laplacian-of-Gaussian implemented in Big-FISH, and the image intensity was manually normalised. Fluorescent signals were then decoded using PixelSpotDecoder in DetectPixels implemented in starfish (https://github.com/spacetx/starfish, *Axelrod et al., 2018*).

## Data analyses and presentation

Microscopy images were presented using the Python package microfilm (Guillaume Witz, Microscopy Imaging Center and Science IT Support, University of Bern, https://github.com/guiwitz/microfilm; *Witz, 2025*; *Figure 1B*). Cell clustering was performed with the method centroid implemented in scipy.cluster.hierarchy.linkage from SciPy (https://scipy.org/; *Figure 2D*). The parameter α in *Equation 5* (*Figure 4C*) and the exponents in *Figure 5C*, *Figure 5—figure supplement 2* were estimated using curve_fit implemented in scipy.optimize from SciPy.

## Acknowledgements

We thank Imaging and Flow Cytometry Core, The Centre for PanorOmic Sciences (CPOS), LKS Faculty of Medicine of The University of Hong Kong for the use of Nikon Ti2-E widefield imaging system; and Central Research Institute of Kawasaki Medical School for the use of Olympus IX83 imaging system. This study was supported by Research Project Grant R04S-003 and R03S-001 from Kawasaki Medical School, The KAWASAKI Foundation of Medical Science and Medical Welfare, and Grant-in-Aid for Scientific Research (C) (22K07091) from Japan Society for the Promotion of Science (JSPS) to MM; Grant-in-Aid for Scientific Research (B) (21H03188) from JSPS to TN; the General Research Fund (17107019) of the Research Grants Council, Hong Kong SAR to HC.

## Additional information

### Funding

| Funder | Grant reference number | Author |
| --- | --- | --- |
| Kawasaki Medical School | R04S-003 | Michi Miura |
| Kawasaki Medical School | R03S-001 | Michi Miura |
| Japan Society for the Promotion of Science | 22K07091 | Michi Miura |
| Japan Society for the Promotion of Science | 21H03188 | Tadasuke Naito |
| Research Grants Council, University Grants Committee | 17107019 | Honglin Chen |

The funders had no role in study design, data collection and interpretation, or the decision to submit the work for publication.

## Author contributions
Michi Miura, Conceptualization, Resources, Data curation, Software, Formal analysis, Supervision, Funding acquisition, Validation, Investigation, Visualization, Methodology, Writing – original draft, Writing – review and editing; Naho Kiuchi, Formal analysis, Validation, Investigation, Visualization, Writing – review and editing; Siu-Ying Lau, Bobo Wing-Yee Mok, Hiroshi Ushirogawa, Resources, Writing – review and editing; Tadasuke Naito, Resources, Funding acquisition, Writing – review and editing; Honglin Chen, Resources, Data curation, Supervision, Funding acquisition, Project administration, Writing – review and editing; Mineki Saito, Resources, Data curation, Supervision, Project administration, Writing – review and editing

## Author ORCIDs
Michi Miura ⓘ https://orcid.org/0000-0002-6943-3782

Reviewer #1 (Public review): https://doi.org/10.7554/eLife.88468.3.sa1
Reviewer #2 (Public review): https://doi.org/10.7554/eLife.88468.3.sa2
Author response https://doi.org/10.7554/eLife.88468.3.sa3

---

# Additional files

## Supplementary files
MDAR checklist

## Data availability
Source data for all the figures are available at the Open Science Framework repository (https://doi.org/10.17605/OSF.IO/H86VM) with CC-By Attribution 4.0 International licence.

The following dataset was generated:

| Author(s) | Year | Dataset title | Dataset URL | Database and Identifier |
|---|---|---|---|---|
| Miura et al. | 2025 | Miura et al eLife-VOR-RA-2023-88468R1 | https://doi.org/10.17605/OSF.IO/H86VM | Open Science Framework, 10.17605/OSF.IO/H86VM |

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
