## [Editor Report · eLife Assessment]

This **important** study combines virology experiments and mathematical modeling to determine the nuclear export rate of each of the eight RNA segments of the influenza A virus, leading to the proposal that a specific retention of mRNA within the nucleus delays the expression of antigenic viral proteins. The proposed model for explaining the differential rate of export is **compelling**, going beyond the state of the art, but the experimental setup is only in partial support and further studies will be needed to confirm the proposed mechanism.

---

## [Referee Report · Reviewer #1 (Public review)]

The authors studied why the two more antigenic proteins of the influenza A virus, hemagglutinin (HA) and neuraminidase (NA), are expressed later during the infection. They set an experimental approach consisting of a 2-hour-long infection at a multiplicity of infection of 2 with the viral strain WSN. They used cells from the lung carcinoma cell line A549. They used the FISH technique to detect the mRNAs in situ and developed an imaging-based assay for mathematically modeling and estimating the nuclear export rate of each of the eight viral segments. They propose that the delay in the expression of HA and NA is based on the retention of their mRNA within the nucleus.

Strength

The study of an unaddressed mechanism in influenza A virus infectious cycle, as is the late expression of HA and NA, by creating a work flow including mRNA detection (FISH) plus mathematical calculations to arrive at a model, which additionally could be useful for general biological processes where transcription occurs in a burst-like manner.

Weakness

The authors built on several assumptions regarding the viral infection to "quantify" the transcript' export rate lacking experimental support. It would greatly improve if more precise experiments could be performed and/or include demonstration of the assumptions made (i.e., empirically demonstrating that cRNA production does not occur within the first 2 hours of infection, and the late expression of HA and NA proteins).

---

## [Referee Report · Reviewer #2 (Public review)]

In this study the authors developed a framework to investigate the export rates of Influenza viral RNAs translocating from the nucleus to the cytoplasm. This model suggests that the influenza virus may control gene expression at the RNA export level, namely, the retention of certain transcripts in the nucleus for longer times, allows the generation of other viral encoded proteins that are exported regularly, and only later on do certain mRNAs get exported. These encode proteins that alert the cell to the presence of viral molecules, hence keeping their emergence to very end, might help the virus to avoid detection as late as possible in the infection cycle.

The study is of limited scope. The notion that some mRNAs are retained in the nucleus after transcription is concluded early on from the FISH data. The model does not contribute much to the understanding and is mostly confirming the FISH data. The export rate is an ambiguous number and this part is not elaborated upon. One is left with more questions since no mechanistic knowledge emerges, and no additional experimentation is attempted to try drive to a deeper understanding.

Comments on revisions:

The authors have implemented the comments that required textual rewriting, which does make the paper clearer. On the experimental side, very little was done. It is fine to answer that the suggested experiments are not relevant or feasible for one reason or another, but one would expect to see some effort in providing other experimental sets to address key comments, and not only to modify a sentence in the text. So in my mind this round of revision feels more like some kind of intellectual discussion, which is fine, but I would have expected more, particularly after so much time has passed. I am still not satisfied with the way the analysis is presented in Fig. 2B, and writing a line about what is not analyzed in the legend, does not seem clear enough.

---

## [Author Response]

The following is the authors’ response to the original reviews.

We thank the editors and reviewers for the comments and suggestions on our manuscript. The main point that we wished to convey in this paper was the concept and the kinetic model that enabled the estimation of nuclear export rate from an image of single mRNAs localised in single cells. By studying the influenza viral transcripts with this model, we report the variation in the mRNA nuclear export rate of the eight viral segments. Of note, the hemagglutinin and neuraminidase mRNAs were the slowest among the eight segments in exiting the nucleus. We agree that the potential mechanism and the biological impact of this observation require further validation, as the reviewers pointed out. We revised our manuscript to describe these points separately (Lines 21-25, Abstract; Lines 86-91, Introduction; Lines 316-320, Results; Lines 372-381, Discussion). We also highlight below, the revisions that we made to address the specific points raised by the reviewers.

Influenza viral transcriptionThe authors used specific settings for their virology experiments and several assumptions regarding their mathematical modelling, so it's extremely important that the reader has the viral life cycle clearly understood before immersing themselves in the results. Thus, a detailed explanation of the viral life cycle, including the kinetics of each step, would be extremely helpful if included in the introduction section. Reviewer #1

We have included the molecular composition of influenza vRNP and the mechanism of viral transcription in the revised manuscript (Lines 46-53).

Line 45: "Eight viral RNA segments are transcribed by the same set of molecular machinery" (Ref. 7). What's known about the arrival of the viral RNA segments in the nucleus? Is it synchronized? The authors will understand that my concern is related to the fact that a differential arrival would indeed impact the transcription and export processes. Reviewer #1

The arrival of eight vRNPs in the nucleus is not synchronised, with each of the eight vRNPs arriving independently (Chou et al. PLOS Pathogens 2013) (Lakadamyali et al, PNAS 2003). This does not compromise our model, as our model estimates the export rate of each mRNA species individually (also please see our response in Model assumption below). This is included in the second paragraph of the Discussion section (Lines 390-400).

Model assumptionEven though I do not have the expertise to assess the authors' mathematical model, I do not doubt its robustness. Even so, I find some virological concerns related to the set-up of their experiments. According to what I understand, the authors performed non-synchronized 2 h-long infections with the WSN strain of influenza A virus. They did this to avoid cRNA production (and cross-reaction of the probes), which they claim to occur "much later than mRNA synthesis". Then they omit the degradation of the mRNAs for their model without giving an explanation for having done so. So, taking all these into account, it seems to me that too many assumptions are made without a strong argument. I understand that they are made in order to simplify their model, but I strongly consider that the model would gain strength if some of these events were experimentally considered. Thus, would it be possible to perform synchronized infections? Would it be possible to empirically demonstrate that cRNA production does not occur within the first 2 hours of infection and/or separate transcription and replication? Would it be possible to incorporate a degradation inhibitor of the mRNAs into their infections? If all these could be achieved, then the results coming out of the mathematical model would be enormously reinforced. Reviewer #1* The study lacks experimental data that would help support the conclusions. For instance, perturbations are many times used to prove a point related to gene expression. An example for Fig. 2 for such an experiment could be to treat the cells with transcription inhibitors (e.g. DRB, 5,6-dichloro1-beta-D-ribofuranosylbenzimidazole). Preventing transcription leaves only mature RNAs in the nucleus, and then using this system one can compare the export rate of different RNAs. Reviewer #2

We agreed that the primary concern in our model was the assumption that the mRNA degradation could be omitted. Synchronised infection is not necessary; in fact, non-synchronised infection is preferred, as we explain later in our response. Additionally, the dominance of mRNA production over the cRNA production has been documented elsewhere. To address mRNA degradation and validate our model estimation, we performed a time-course measurement using baloxavir. Baloxavir efficiently blocks the viral transcription by inhibiting the nuclease activity in PA. DRB, suggested by the reviewer, allows influenza viral transcription and causes viral transcripts to accumulate in the nucleus for unknown mechanisms (Amorim et al. Traffic 2007 and our observation using smFISH, not shown). The additional experiment, now presented in Fig. 5 in the revised manuscript, indicated that the mRNA degradation is minimal, and the export rate estimated in our model and the time-course experiment agreed well for the HA segment. The experiment raised the possibility that the time-course measurement underestimates the export rate of transcripts that exit the nucleus rapidly, such as NP. A real-time imaging of single transcripts would be necessary to directly measure the true nuclear export rate; however, this is beyond the scope of our paper. The new result is now presented in Fig. 5, Supplementary figures 3 and 4, and in the main text (Lines 322-360). An alteration was also made in Line 286 to guide to Fig. 5. The Materials and Methods section was updated (Lines 478-482).

We note that our model does not require synchronised infection. Even under synchronised infection, such as incubating cells with the virus at 4°C to facilitate attachment and subsequently shifting to 37°C to allow viral entry, the inherent heterogeneity in vRNP migration to the nucleus still remains. This randomness does not compromise our model; rather, our model exploits this random arrival of each vRNP in each cell in the system. This variation, in turn, generates cells carrying varying amounts of transcripts, enabling the estimation of nuclear export rate. Importantly, more variation ensures the broader distribution of transcript levels, enabling more precise parameter fitting in our model. It is also important to note that our model does not require the correlation between segments. Our model estimates the export rate of each mRNA species individually. These important points were explained in the Discussion section (Lines 390-400).

* There is no concrete value given for the export rates and what they might mean biologically (e.g. time present/stuck in the nucleus) - Fig. 4D. This leaves the reader in the dark. Reviewer #2

The export rate lambda (previously denoted as *k*) in our model (Fig. 4) and the decay constant *k* in the time-course measurement (Fig. 5) represent the proportion of mRNAs exported from the nucleus in an infinitesimal time, defining the nuclear export rate. This has been clarified in the revised manuscript (Lines 314-316), with some alterations to make the parameter use more comprehensive.

- The Greek letter *k* previously used in Fig. 4 and the associated equations was consistently replaced with lambda to avoid the confusion with the parameter *k* that is subsequently used for the exponent decay in Fig. 5 in the revised manuscript.

- The Greek letter epsilon (previously used to represent *export*) was replaced with mu, slightly more common for representing the rate of transport.

- The term “velocity” was consistently replaced with “rate” in the context of the nuclear export (Lines 163, 215, 320, 441).

- The phrase “molar concentrations of mRNAs” was corrected for “molecules of mRNAs” (Line 282).

Also, we have now described our model in two sections: “Conceiving the model” and “Implementing a kinetic model to estimate the nuclear export rate” in the Result. The first section outlines the conceptual framework of the model, and the second focuses on its implementation and the parameter extraction (Lines 227 and 277).

Applicability of the modelLines 27-29. "Our framework presented in this study can be widely used for investigating the nuclear retention of nascent transcripts produced in a transcription burst." In my opinion, this is the strongest point of the manuscript: developing a mathematical model to analyze nuclear export retention as a mechanism of protein expression control, which could lay the foundation for further biological processes. The authors revisit this idea in the Discussion section. However, which would be those processes for which the model could be helpful? I consider that a more conspicuous discussion on this topic would broaden the readers scope, a crucial point under the eLife scope. Reviewer #1* Could this framework be used to quantify the nuclear export rate of cellular RNAs? According to the explanation in the Discussion, it would seem that this approach is limited to quantifying the export rate of influenza RNAs. Reviewer #2

Our model is not limited to the influenza virus infection. Our model is applicable for systems where transcription is initiated concurrently, such as when stimuli trigger the activation of a certain set of genes for transcription. Therefore, this makes it particularly valuable for quantifying the nuclear retention of mRNAs in a transcription burst. This point is reiterated in Line 383-390.

Potential mechanisms for differential nuclear export rate of viral segments* There is no mechanistic insight in the study. The idea driven by this study is that gene expression is regulated by the RNA export rate. But how is that explained? Is there any molecular pathway or explanation for this model? If the transcripts are ready for export, why do the mRNAs stay inside the nucleus? One option to consider are the export factors. Viral RNAs are exported by different pathways as mentioned (line 362), or by TREX2 (Bhat P et al Nat Comm 2023). The data shows that there is no difference observed in the export rate of different pathways. How about knocking down an important export factor to show how this affects the export rates. Or the opposite, overexpress a certain factor, would this change the nucleus/cytoplasm distribution of the retained RNAs. Reviewer #2

As we discussed in the paper, we are beginning to consider that each viral segment has an intrinsic sequence that determines its nuclear export rate, because previous studies on the export factors does not fully explain the variation in the nuclear export rate observed in our study. As the reviewer suggested, a recent study (Bhat et al. Nature Communications 2023) exactly pointed out the internal sequence in the HA segment, aligning with our working hypothesis. This point is discussed and their work (Bhat et al. 2023) has been cited in the Discussion section in the revised manuscript (Lines 446-449).

Biological impact of the nuclear retentionThe authors mention several times throughout the manuscript that the virus might use the nuclear retention of mRNA for HA and NA to postpone the expression of these antigenic molecules. At this point, I need to admit that a great question mark appeared in my mind, maybe related to the fact that some knowledge is lacking in my analysis. Lines 328-330: "On the other hand, pushing back the expression of viral antigens HA and NA would be beneficial for the virus to delay the host immune response against the infected cells in which the virus is being replicated." As I tend to understand, the host immune response recognizes HA and NA within the viral particle, if so and independently of the time that HA and Na arrive at the virus assembly step, the progeny' viral particles that are complete and extruded from the cells would be those awakening the host immunity response. If this is right, how would the delayed export of those proteins from the nucleus (and their late expression) be beneficial for delaying the immune response? I would appreciate an explanation for this point, and if I am wrong, then there could exist a relationship between nuclear export rate and the pathogenicity of different strains of influenza A virus. If so, could the authors challenge their model with additional viral strains showing a differential immune response pattern? A deeper analysis in this direction would greatly strengthen the message in their manuscript. Reviewer #1* Is the timing of viral protein appearance in accordance with the time the mRNA is exported to the cytoplasm. It is logical that the first mRNA to go to the cytoplasm would be the first to become a protein. Can the authors show that nuclear retention of mRNA would push back the expression of the viral antigens HA and NA. Reviewer #2

Three types of immune reactions are being studied extensively. The first is the humoral immune response, where antibodies target the viral antigens HA and NA on the viral envelope, coating and inactivating the viral particles. The second is the cytotoxic T cell response. There is growing evidence that cytotoxic T cells react against NP, eliciting cross-reaction to broader range of influenza viral strains. This reaction is not specific to HA and NA, and antigens are processed in the cytoplasm and presented on the MHC. The third is antibody-dependent cellular cytotoxicity (ADCC), where antibodies recognise the viral proteins on the cellular surface (HA and NA) of infected cells, facilitating their elimination by the NK cells. Although protein translation may begin as soon as the first mRNA exits the nucleus, the virus may delay the peak of the antigen production and therefore, postpone the NK-mediated ADCC. This specific point, along with references to ADCC in influenza virus infection, has been clarified in the Discussion section (Lines 377-381).

Data analysis and presentationLines 99-101. "Viral mRNAs were detected as single diffraction-limited spots in the three-dimensional image stacks, allowing for absolute mRNA quantification (Fig. 1B)". What do the authors mean to say by "absolute mRNA quantification"? Do they refer to the total spots or the total mRNAs? Is it assumed that one spot corresponds to a single mRNA transcript? This is not clear at all for this reviewer, which could be the situation for a potential reader. Since it's the beginning of the story, this should be clearly stated in the manuscript. Reviewer #1

Each spot of fluorescent signal corresponds to a single molecule of viral mRNA. We quantified the absolute number of transcripts in each cell. This is clarified in the revised manuscript (Lines 104-106).

* Line 151: does the baseline change according to the RNA in question? The authors say that the "baseline is defined by the median of the Z distribution of peripheral mRNAs" - it seems that the number 0.731 refers only to one type of RNA (which is not mentioned at all not in the text and not in the legend). Reviewer #2

The baseline was set using the NP mRNAs in the cytoplasm because the NP mRNA showed the widest distribution across the cytoplasm (Line 157).

* Also, what is all the signal that is seen outside the marked cells in Fig. 2B? There seems to be significant background in the field, does this mean much false-positive in the multiplex FISH? If so, then how do the authors know that the staining inside the cells isn't to some degree non-specific? It would be necessary to back this up with some other type of quantitative assay like qRT-PCR. Reviewer #2

The cells were removed from the analysis if the cytoplasmic boundary touched any edge of the field-of-view, while the signals were recovered across the entire field-of-view. This is clarified in the figure legend (Lines 194-195).

Others* The meaning and explanation for Figure 1H -are unclear. Rephrase and make the legend more reader friendly. Reviewer #2

We made alterations to the legend (Lines 132-134) and the relevant lines in the main text (Lines 148-151).

* Fig. 2E: Is this the total transcript count or only in the nucleus? Would it be possible to find some correlation between the segments if a pair-wise analysis is performed according to nuclear-cytoplasm distribution? Reviewer #2

The total counts are presented. This is clarified in the legend (Lines 199-200).

* Abstract -"A mathematical modelling indicated that the relationship between the nuclear ratio and the total count of mRNAs in single cells is dictated by a proxy for the nuclear export rate." - this sentence is very unclear. Reviewer #2

The sentence was removed in the revised manuscript (Line 21). This removal did not affect the overall meaning in the abstract. We also made an alteration to Line 279 that contained a similar phrase.

* The use of the word "acutely" (lines 16 and 35) is strange. Reviewer #2

They have been removed (now Lines 15, 33).

* Line 157 - "This result indicates that the velocity of viral mRNA export from the nucleus varies according to the viral segments." - not velocity, maybe timing. Reviewer #2

We consistently replaced “velocity” with “rate” (Lines 163, 215, 320, 441).

* Reference for line 41. Reviewer #2

A reference (Waker et al. Trends Microbiol. 2019) has been cited (Line 39).

* Reference for lines 105-106. Reviewer #2

The gene length of each segment was indicated in the sentence (Line 137).

* Line 264- why here is 0.02 M.O.I used compared to line 97 where 2 is used? Reviewer #2

We used M.O.I. of 0.02 to allow for spot quantification over longer periods of observation (Lines 269-270).

* NS1 is expressed at late infection times and might alter the nuclear export of viral mRNAs (line 352). Need to show that indeed it is not expressed in the experiments done here. Reviewer #2

It is not possible to definitely prove that NS1 is not expressed due to the sensitivity limitations. However, we minimised the its impact by investigating at the early time point (Lines 415416).

* Line 459- 30% formamide? Is this correct or should it be 10%? Reviewer #2

This is correct. The probes used were longer than the others for smFISH. Therefore, we washed away the probes with the stringent condition.